# A Novel Quadruple Gene-Deleted BoHV-1-Vectored RVFV Subunit Vaccine Induces Humoral and Cell-Mediated Immune Response against Rift Valley Fever in Calves

**DOI:** 10.3390/v15112183

**Published:** 2023-10-30

**Authors:** Selvaraj Pavulraj, Rhett W. Stout, Elise D. Barras, Daniel B. Paulsen, Shafiqul I. Chowdhury

**Affiliations:** Department of Pathobiological Sciences, School of Veterinary Medicine, Louisiana State University, Baton Rouge, LA 70803, USA; pselvaraj1@lsu.edu (S.P.); rstout1@lsu.edu (R.W.S.); e.barras@ufl.edu (E.D.B.); dpauls1@lsu.edu (D.B.P.)

**Keywords:** BoHV-1 mutant, BoHV-1 vector, subunit-vaccine, RVFV, immunogenicity, Gn and Gc, cattle

## Abstract

Rift Valley fever virus (RVFV) is considered to be a high biodefense priority based on its threat to livestock and its ability to cause human hemorrhagic fever. RVFV-infected livestock are also a significant risk factor for human infection by direct contact with contaminated blood, tissues, and aborted fetal materials. Therefore, livestock vaccination in the affected regions has the direct dual benefit and one-health approach of protecting the lives of millions of animals and eliminating the risk of severe and sometimes lethal human Rift Valley fever (RVF) disease. Recently, we have developed a bovine herpesvirus type 1 (BoHV-1) quadruple gene mutant virus (BoHV-1qmv) vector that lacks virulence and immunosuppressive properties due to the deletion of envelope proteins UL49.5, glycoprotein G (gG), gE cytoplasmic tail, and US9 coding sequences. In the current study, we engineered the BoHV-1qmv further by incorporating a chimeric gene sequence to express a proteolytically cleavable polyprotein: RVFV envelope proteins Gn ectodomain sequence fused with bovine granulocyte-macrophage colony-stimulating factor (GMCSF) and Gc, resulting in a live BoHV-1qmv-vectored subunit vaccine against RVFV for livestock. In vitro, the resulting recombinant virus, BoHV-1qmv Sub-RVFV, was replicated in cell culture with high titers. The chimeric Gn-GMCSF and Gc proteins expressed by the vaccine virus formed the Gn–Gc complex. In calves, the BoHV-1qmv Sub-RVFV vaccination was safe and induced moderate levels of the RVFV vaccine strain, MP12-specific neutralizing antibody titers. Additionally, the peripheral blood mononuclear cells from the vaccinated calves had six-fold increased levels of interferon-gamma transcription compared with that of the BoHV-1qmv (vector)-vaccinated calves when stimulated with heat-inactivated MP12 antigen in vitro. Based on these findings, we believe that a single dose of BoHV-1qmv Sub-RVFV vaccine generated a protective RVFV-MP12-specific humoral and cellular immune response. Therefore, the BoHV-1qmv sub-RVFV can potentially be a protective subunit vaccine for cattle against RVFV.

## 1. Introduction

Rift Valley fever (RVF) is a mosquito-borne zoonotic viral disease of cattle, sheep, and goats caused by the Rift Valley fever virus (RVFV). The RVFV is an emerging pathogen that maintains high biodefense priority based on its threat to livestock, resulting in high mortality in newborn animals and mass abortion upon infecting pregnant animals. The virus causes hemorrhagic fever in humans [1,2,3]. RVFV-infected animals serve as the source of human infections.

RVFV belongs to the genus Phlebovirus, family Bunyaviridae, and has a negative-stranded RNA, consisting of S-, M-, and L-segments. The S-segment (1690 nucleotides; nt) expresses N protein and nonstructural protein S (NSs) in an ambi-sense manner. The M-segment (3885 nt) encodes NSm, amino-terminal glycoprotein (Gn), and carboxyterminal glycoprotein (Gc) in a single open reading frame (ORF) but is cleaved proteolytically in the endoplasmic reticulum (ER) and co- and post-translationally into Gn and Gc, which also form heterodimeric complex in the ER, concurrently [4]. The L-segment (6404 nt) encodes L protein, a viral RNA-dependent RNA polymerase. Both N and L proteins are required for viral replication and transcription. The Gn and Gc heterodimer complex formation is required for Gn and Gc transport to and maturation in the Golgi, and Gn–Gc envelope incorporation [4]. Together, they are also the key target of RVFV-specific neutralizing antibodies [5] and the CD4 positive memory T-cells that induce the RVFV-specific recall-neutralizing IgG response [6].

RVFV is widely distributed in sub-Saharan Africa, with epizootic activity affecting animals in Kenya, Tanzania, Zambia, and Uganda [7]. Rapid intercontinental commerce and a lack of effective control measures threaten to expand the geographic range of RVFV. A recent example is an expansion of RVFV to the Arabian Peninsula [8]. Therefore, the availability of an efficacious vaccine against RVFV will be exceedingly valuable to protect the U.S. livestock population and ultimately prevent transmission to humans if the RVFV is introduced accidentally or through an agroterrorism event.

A live-attenuated RVFV MP12 vaccine was developed from the virulent ZH-548 strain [9,10,11]. The vaccine retained residual virulence but generated a neutralizing antibody response in cattle, sheep, monkeys, and humans. The vaccine can induce abortion in 4% of ewes and teratogenic effects in 14% of newborn lambs [5,12,13]. In addition, under field conditions, there is the potential for MP12 to regain virulence or revert to wild-type (wt) due to reassortment with circulating wt strains [14,15].

Several live virally vectored subunit-RVFV vaccines, using replication-competent, i.e., Newcastle disease virus [16] and Capripox virus [17], and replication-defective adenovirus [18,19] and vaccinia virus Ankara (MVA) [20] have been developed. Further, baculovirus-expressed RVFV antigen-based subunit vaccines [21] and RVF virus-like particle vaccines [22] have also been developed and tested for vaccine efficacy against RVFV. However, each has its own deficiency concerning vaccine efficacy, safety, or booster requirements [13].

Recently, our lab constructed a BoHV-1 quadruple gene-mutated virus (BoHV-1qmv) vaccine vector in which we deleted the genes associated with virulence, immune evasion, and those required for the nasal virus shedding after latency reactivation (due to the defective anterograde transport from the neuron cell bodies in the trigeminal ganglia (TG) to axon termini in the nasal epithelium), i.e., glycoprotein E cytoplasmic tail (gE-CT), US9, UL49.5, and envelope glycoprotein G (gG) [23,24,25]. Upon intranasal vaccination, the BoHV-1qmv vector replicated moderately in the nasal epithelium, lacked virulence in calves, and did not immunosuppress [24]. Notably, the BoHV-1qmv has the serological marker distinguishing vaccinated animals from wt-infected animals (DIVA property). We have used this BoHV-1qmv as a vector to incorporate bovine viral diarrhea virus type 2 (BVDV-2) envelope proteins E2 and chimeric Erns fused with bovine granulocyte-macrophage colony-stimulating factor (GMCSF). The resulting BoHV-1qmv sub-BVDV2 protected the vaccinated calves against the virulent BVDV-2 challenge [23].

In this study, we have constructed a BoHV-1qmv-vectored subunit vaccine against RVFV by incorporating the chimeric RVFV Gn-GMCSF (Gn fused with bovine GMCSF) and RVFV Gc proteins in the gG-deletion locus. Further, we tested its safety and vaccine efficacy in calves for the protective neutralizing antibody and interferon-gamma (IFN-γ)-producing peripheral blood mononuclear cells (PBMCs; cellular immune response). The results showed that BoHV-1qmv Sub-RVFV is highly safe for vaccination in calves, and a single dose of the prototype vaccine resulted in moderate levels of RVFV-specific serum-neutralizing antibody titers and RVFV-specific cell-mediated immune response.

## 2. Materials and Methods

### 2.1. Cells and Medium

The bovine esophagus cells (KOP-R cells, CCLV-RIV 244) were kindly provided by Dr. Thomas Mettenleiter (Collection of Cell Lines in Veterinary Medicine, Friedrich-Loeffler Institute, Insel Riems, Germany). The KOP-R, 293T, and the Madin Darby bovine kidney (MDBK) cell lines were maintained in Dulbecco’s modified Eagles medium (DMEM #10-017-CV, Corning^®^, Corning, NY, USA) supplemented with 10% heat-inactivated EquaFETAL serum (Atlas Biologicals, Fort Collins, CO, USA) and 1× antibiotic/antimycotic solution (#30-004-CI; Corning^®^). The PBMCs collected from the calves for the cell-mediated immune response assay were plated in complete RPMI medium (#11875085, Gibco™, Waltham, MA, USA) supplemented with heat-inactivated 10% Equa-FETAL serum, 2-Mercaptoethanol 50 µM, L-glutamine 20 mM, HEPES 25 mM and 1× antibiotic/antimycotic solution.

### 2.2. Viruses

The BoHV-1 wt Cooper (Colorado-1) strain was obtained from the American Type Culture Collection (#VR-864, ATCC^®^, Manassas, VA, USA), and low-passage viral stocks were maintained at −80 °C. BoHV-1qmv has been constructed and characterized previously [23]. The BoHV-1 wt and the BoHV-1 recombinant viruses were titrated by plaque assay in MDBK cells, as described previously [23]. The attenuated RVFV vaccine strain, MP-12, was obtained from Dr. Chris Mores (The George Washington University). Low-passaged MP12 stocks were propagated and titrated in KOP-R cells.

#### 2.2.1. Commercial Antibodies

Mouse anti-V5 mAb (#R960-25, Thermo Fisher Scientific^®^, Waltham, MA, USA), anti-FLAG mouse mAb (#F1804, Sigma-Aldrich^®^, St. Louis, MO, USA), and anti-FLAG rabbit antibody (#F7425, Sigma-Aldrich^®^) were purchased. Secondary antibodies, donkey anti-mouse IgG Alexa Fluor 488 (#A-21202), goat anti-mouse IgG horseradish peroxidase (HRP) (#32430), donkey anti-rabbit IgG Alexa Fluor 647 (#A31573, and donkey anti-rabbit HRP conjugate (#31458) were purchased from Thermo Fisher scientific^®^.

#### 2.2.2. Rabbit Anti-RVFV Gn and Gc Polyclonal Antibodies Production

Anti-RVFV Gn and Gc rabbit polyclonal antibodies were generated commercially (BioMatik, Kitchener, ON, Canada) against a RVFV Gn- and Gc-specific peptides, respectively (Table 1).

### 2.3. Construction of BoHV-1qmv Vector Virus Expressing the Chimeric RVFV Gn-GMCSF and Gc Proteins

#### 2.3.1. Construction of RVFV Gn-GMCSF-Gc Expression Cassette

The plasmid pgGΔ (Figure 1) was constructed previously [23]. Briefly, in the plasmid clone pgGΔ, the first 67 aa coding nucleotides of the gG ORF were deleted, and the KpnI and HindIII sites were incorporated into the gG-deletion locus. Further, the gG-deletion locus is flanked by gG upstream 1 kb on the left, within the NotI-KpnI, and on the right, within the 1.16 kb HindIII-NsiI bracketed sequence, containing the partial gG carboxy-terminal and gG-gD intergenic sequences (Figure 1). In this configuration, the remaining gG carboxy-terminal sequences would not be translated, and consequently the gG gene is inactivated.

To incorporate a chimeric RVFV Gn–Gc gene sequence, we first designed a gene cassette that included, from 5′ to 3′, a KpnI restriction site, cytomegalovirus immediate-early gene (CMV-IE) promoter sequence (GenBank accession #U55763; nt 1-605), a Kozak sequence, a BoHV-1 gD signal sequence (nt 118,819 to 118875, #JX898220; aa 1–19, #AFB76672.1), and a nucleotide sequence for RVFV Gn with cytoplasmic tail (GenBank accession #ABD38819.1; aa 1, 154-581 and 605-674; lacking transmembrane domain), followed by a nucleotide sequence of bovine GMCSF (GenBank accession #AAA66075.1; aa 1, 18-143; lacking signal sequence) fused in frame with the C-terminal of RVFV Gn, a nucleotide sequence for a FLAG tag, peptide 2A (P2A) sequence with a GSG aa [26] to improve cleavage efficiency (GSG-ATNFSLLKQAGDVEENPGP), a nucleotide sequence for RVFV Gc with transmembrane and cytoplasmic domain (GenBank accession #ABD38819.1; aa 1, 691-1197), a nucleotide sequence of V5 epitope tag, stop codon (TAA), the SV40 Poly A site (GenBank accession #U55763; nt 1411-1640), and a restriction site for the HindIII (Figure 1 and Appendix A). The RVFV Gn, Gc, and bovine GMCSF sequences were codon optimized for *Bos taurus* (Biomatik), and the entire 4.459 kb chimeric gene sequence (RVFV Gn-GMCSF-FLAG-P2A-Gc-V5- Sv40 PolyA) was synthesized and cloned into the KpnI-HindIII sites of the pUC57-Bsa1-Free plasmid (Biomatik).

The 4.459 kb KpnI-HindIII fragment containing the entire chimeric gene was then inserted into the KpnI-HindIII sites of the plasmid pgGΔ. The nucleotide sequence of the resulting plasmid pBoHV-1 gGΔ/RVFV Gn-GMCSF-P2A-Gc-INS (designated hereafter as pgGΔ-Gn-Gc+) was verified. Further, the expression of the chimeric Gn-GMCSF and Gc proteins upon P2A cleavage was validated by transfection of the pgGΔ-Gn-Gc plasmid DNA in the KOP-R cells, followed by SDS-PAGE and immunoblotting analyses with the RVFV Gn, RVFV Gc, and FLAG tag- and V5-tag-specific antibodies.

#### 2.3.2. Construction of BoHV-1qmv Expressing RVFV Gn-GMCSF-Gc Expression Cassette (BoHV-1qmv Sub-RVFV)

To construct the BoHV-1qmv Sub-RVFV, the linearized (NotI or NsiI digested) pgGΔ-Gn-Gc DNA was co-transfected along with the full-length BoHV-1qmv DNA into the 293T cells, as described previously [23]. Several putative BoHV-1qmv Sub RVFV recombinant viral plaques were purified and first verified by nucleotide sequence analysis to incorporate the chimeric RVFV Gn-GMCSF and Gc gene sequences. The expression of the chimeric proteins was verified by immunoblotting analysis with FLAG and V5 tags (Section 2.3.1) or the rabbit polyclonal Gc and Gn-specific antibodies generated above (Section 2.3.2). One representative BoHV-1qmv Sub-RVFV recombinant virus was selected for virus stock preparation and further characterization.

### 2.4. Growth Kinetics and Plaque Size Assay

The growth kinetics of BoHV-1qmv Sub-RVFV was evaluated and compared with that of the BoHV-1 wt by standard one-step growth kinetics assay, as described previously [23]. To determine the cell-to-cell spread property of BoHV-1qmv Sub-RVFV compared with that of BoHV-1 wt, average plaque sizes of wt and BoHV-1qmv Sub-RVFV viruses were determined by measuring approximately 150 randomly selected plaques of each virus under an inverted fluorescent microscope (Olympus IX71, Shinjuku City, Tokyo, Japan) [23]. The actual plaque diameters were measured using ImageJ^®^ software version 1.53t (National Institute of Health, Bethesda, MD, USA), as previously described [23].

### 2.5. Characterization of RVFV Gn and Gc Proteins

For Western blot analysis of chimeric RVFV Gn-GMCSF and Gc expression by BoHV-1qmv Sub-RVFV, KOP-R cells were infected with BoHV-1qmv Sub-RVFV and RVFV vaccine strain MP12. When the cytopathic effects (CPE) were 80–90% (48-hr post-infection for BoHV-1qmv Sub-RVFV and 72-hr post-infection for MP12), cells were harvested and processed for cell lysates, as described previously [23]. The solubilized proteins were aliquoted and stored at −80 °C. SDS-PAGE and Western immunoblotting analysis were performed to validate the chimeric Gn-GMCSF and Gc protein expression by the BoHV-1qmv Sub-RVFV, as previously [23]. MP12 lysates served as a positive control for Gn and Gc.

### 2.6. Immunoprecipitation, Endoglycosidase H (Endo H), and Peptide-N-Glycosidase F (PNGase-F) Sensitivity Analyses

Mock- or virus-infected KOP-R cell lysates were immunoprecipitated for the analysis of, Gn–Gc complex formation and for testing the Endo H and PNGase F sensitivity as described previously [27]; however, in this study we used protein G Sepharose beads (Protein G Sepharose^®^ 4 Fast Flow, # GE17-0618-01, Cytiva™, Marlborough, MA, USA) instead of protein A Sepharose beads. For Endo H and PNGase F digestion of the immunoprecipitated protein, we followed a similar protocol as described previously. The immunoprecipitated and/or Endo H/PNGase F-digested proteins were subjected to SDS-PAGE and immunoblotting with FLAG tag/Gn (Gn-specific) or V5 tag (Gc-specific) antibodies.

### 2.7. Vaccine Virus Stability In Vitro

After ten continuous passages in cell culture and one in calf, BoHV-1qmv Sub-RVFV genomic stability was determined by nucleotide sequencing. Further, the expression of the Chimeric RVFV proteins was verified by immunofluorescence assays with FLAG tag (Gn-specific) and V5 tag (Gc-specific) antibodies.

### 2.8. Calves and Experimental Design

Animal infection, handling, sample collection, and euthanasia protocols were previously approved by the LSU Institutional Animal Care and Use Committee (Protocol #20-028). Eleven six-month-old crossbred steer, bull, or heifer non-vaccinated calves were obtained from a BVDV-free supplier. The calves were pre-tested for BoHV–1 serum neutralizing (SN) antibody titers and bovine viral diarrhea virus (BVDV) viremia (VetMAX™-Gold BVDV PI Detection Kit, #4413938, Thermo Fisher Scientific, Plaquemine, LA, USA) to ensure BoHV–1/BVDV-free status. Calves with <4 BoHV-1-specific serum-neutralizing (SN) antibody titers and negative for BVDV viremia were selected for the study. Calves were randomly divided into two groups. Group 1 had three calves and Group 2 had eight calves. The two groups were housed in pens well separated from each other (by more than 100 feet) and located in the School of Veterinary Medicine large animal isolation barn. Foot baths were located at the main entry and in front of each pen.

Vaccination, challenge, and sample collection schemes are shown in Figure 2. After a week of acclimatization, the vector group (Group 1) was given, intranasally (IN), BoHV-1qmv vector 5 × 10^7^ PFU/nostril (a total of 1 × 10^8^ PFUs per animal) and, subcutaneously (SQ), 5 × 10^7^ PFU. Similarly, the calves in the BoHV-1qmv Sub-RVFV vaccine group (prototype) vaccine group (Group 2) were given, IN, BoHV-1qmv Sub-RVFV 5 × 10^7^ PFU/nostril (a total of 1 × 10^8^ PFUs per animal) and, SQ, 5 × 10^7^ PFU. Blood for serum and PBMCs and nasal swabs were collected at days 0, 3, 5, 7, 10, 14, 21, 28, and 33, post-vaccinations (dpv) (Figure 2). The experiment was terminated at 33 dpv and calves were euthanized with Euthasol^®^ (Euthanasia Solution; pentobarbital sodium and phenytoin sodium).

### 2.9. Clinical Examination of Calves Following Immunization

Calves were clinically assessed for rectal temperature, feed, and water intake, from 0 to 33 dpv (until euthanasia). The clinical assessment included rectal temperature and nasal and ocular discharges.

### 2.10. Collection and Processing of Samples from Immunized Calves

The EDTA–blood (purple top), serum (red top), and nasal swab samples were collected as shown in Figure 2 and processed for PBMCs, serum collection, and virus isolation, as previously described [23]. Nasal swabs were collected in 1 mL of cell culture media, supplemented with 2× antibiotic/antimycotic solution. Vaccine virus/vector virus nasal virus shedding was determined by plaque assay and qPCR analysis [23].

### 2.11. Serum Virus Neutralization Assay for BoHV-1 and RVFV Vaccine Strain MP-12 by Plaque Reduction Assay for BoHV-1 and RVFV

Heat-inactivated sera samples (56 °C for 30 min) were used for the BoHV-1- and RVFV-specific plaque reduction assay. The BoHV-1-specific plaque reduction assay was described previously [23]. The procedures for the MP-12-specific plaque reduction assay was similar to the above; however, KOP-R cells were used instead of MDBK cells and the titration plates were fixed with 10% formaldehyde at 72 hrs post-infection.

### 2.12. DNA Isolation and Quantitative PCR (qPCR)

To quantify the BoHV-1qmv vector and BoHV-1qmv Sub-RVFV genome copies in the nasal swabs of the vaccinated calves, total DNA was isolated using a QIAamp^®^ DNA mini kit (#51306, Qiagen, Hilden, North Rhine-Westphalia, Germany). BoHV-1 genome copies were determined by TaqMan probe-based real-time qPCR in an ABI PRISM™ 7900HT Sequence Detection System (Applied Biosystems, Waltham, MA, USA), targeting major capsid protein (VP5) ORF coding sequence primer pairs (Table 2). Each time, the PCR reaction setup was run with six standards of known quantity (10^1^ to 10^6^ copies per reaction). BoHV-1 genome copies in the nasal samples (normalized to 100 ng of total DNA) were compared with the generated standard curves. The assay was duplicated, and results were expressed as BoHV-1 genome copies per 100 ng of DNA.

### 2.13. RVFV-Specific Cell-Mediated Immune Response Assay

Aliquots of PBMCs collected from the immunized calves on days 0, 14, and 21 post-vaccination and stored in liquid nitrogen were thawed and seeded at the rate of 0.5 × 10^6^ cells/well in a 96-well plate (100 µL volume). After an overnight incubation in RPMI medium at 37 °C, PBMCs were stimulated with 10 µg/mL of heat-inactivated RVFV antigen. Concanavalin A (ConA; 5 µg/mL) served as a positive control, and plain RPMI medium was used as a negative control. At day 3 post-antigen stimulation, PBMCs were restimulated with 5 µg/mL RVFV antigen. At day 4 post-antigen stimulation, PBMCs were harvested and subjected to total RNA isolation. Total RNA was isolated using an RNeasy Mini Kit (#74104, Qiagen), as per the manufacturer’s instructions. A cDNA synthesis was performed with a Verso cDNA Synthesis Kit (AB-1453/B; Thermo Fisher Scientific^®^, Plaquemine, LA, USA). IFN-γ mRNA transcript Ct values and copies were determined by real-time qPCR using the primers and probes listed in Table 2 (IFN-γ; GenBank accession # XM_027543610.1) and normalized to a standard curve generated with the host-specific bovine housekeeping gene, Glyceraldehyde 3-phosphate dehydrogenase (GAPDH; GenBank accession #XM_ 027541122.1). The assay was performed in duplicate. The A 2^−∆∆Ct^ method was used to calculate the relative fold-change in the INF-γ expressions between the RVFV antigen-stimulated and unstimulated PBMCs.

### 2.14. Indirect Immunofluorescence Assay

Indirect immunofluorescence assay (IIFA) was performed, as described previously [28].

### 2.15. Statistical Analysis

Statistical analyses were performed using GraphPad PRISM^®^ 5.01 software (San Diego, CA, USA). Normally distributed group samples were analyzed with a one-way ANOVA test followed by a multiple comparisons test. A ‘*p*’ value of less than 0.05 was considered significant for all analyses.

## 3. Results

### 3.1. Characterization of BoHV-1qmv Sub-RVFV Recombinant Virus

#### 3.1.1. BoHV-1qmv Sub-RVFV Virus Expresses the Chimeric RVFV Gn-GMCSF-FLAG and Gc-V5 Proteins

Sequence analyses of the BoHV-1qmv Sub-RVFV genomic region spanning the Gn-GMCSF, P2A, and Gc chimeric genes and its BoHV-1-specific flanking sequences (Figure 1, approx. 1000 bp on each side) validated the integrity and their appropriate insertion at the gG deletion locus. Further, the expression of chimeric FLAG-tagged Gn-GMCSF and V5-tagged Gc proteins in the BoHV-1qmv Sub-RVFV-infected KOP-R cell lysates were verified in comparison to the MP12-expressed Gn and Gc proteins by SDS–PAGE/Western immunoblotting. As depicted in Figure 3, the RVFV Gn-specific and FLAG-specific antibodies detected a protein band with a molecular mass of approximately 72 kD in the case of BoHV-1qmv Sub-RVFV-infected cell lysates (Figure 3). However, the same Gn-specific rabbit antibody detected a protein band of approx. 60 kD in the case of RVFV MP12-infected cell lysates (Figure 3). The estimated molecular mass (https://web.expasy.org/compute_pi/, accessed on 24 March 2023) of the chimeric FLAG-tagged Gn-GMCSF protein and the MP12 Gn are 70.1 kD (RVFV Gn—54.8 kD; bovine GMCSF—14.3 kD; FLAG tag—1 kD) and 58.8 kD, respectively, which is in both cases 2 kD more than their estimated sizes. The Gn ORF contains one N-linked glycosylation site (N438) [29]. Therefore, these results are consistent with the increased molecular mass of the chimeric Gn-GMCSF and the native Gn (MP12).

The RVFV Gc- and V5-tag-specific antibodies detected an approximately 62-kD band in the BoHV-1qmv Sub-RVFV-infected KOP-R cell lysates. At the same time, the RVFV Gc-specific antibody detected the 61-kD band in the RVFV MP 12-infected KOP-R lysate. The estimated molecular mass of RVFV Gc-V5 chimeric protein is 56.8 kD (55.44 kD for Gc and 1.4 kD for V5). The Gc ORF sequence contains four N-glycosylation sites (N794, N829, N1035, and N1077) out of which one (N829) was reported not to be glycosylated [29]. The increase (approximately 5 kD) in the molecular mass of Gc-V5 and native Gc is consistent with the glycosylation at the three reported sites.

#### 3.1.2. Glycosylation of Chimeric Gn-GMSCF and Gc

As noted above, Gn and Gc contain one and three functional glycosylation sites, respectively [29]. The same study also reported that both Gn and Gc of the MP12 strain are PNGase F- and Endo H-sensitive. Generally, the glycoprotein obtains its high mannose sugars co-translationally in the ER, and the complex sugars are added to the protein when the protein is processed further post-translationally in the Golgi complex. Both the ER and Golgi processed proteins are PNGase F-sensitive, but only the Golgi-processed complex sugar-containing proteins are Endo H-resistant. Previously, it was reported that, in MP12, both Gn and Gc are sensitive to PNGase F and Endo H, indicating that they are not processed post-translationally in the Golgi [29].

To determine the chimeric Gn-GMCSF and Gc processing post-translationally, we performed immunoprecipitation followed by PNGase F and Endo H digestions, individually, as described in Section 2. For this, approx. 250 μg of BoHV-1qmv Sub-RVFV-infected cell lysates were either immunoprecipitated with anti-Gn or anti-Gc antibodies; then, the immunoprecipitated proteins were either treated with PNGase F or Endo H or untreated. Finally, the digested proteins were analyzed by SDS-PAGE and immunoblotting with anti-FLAG- and anti-V5-specific mAbs, respectively. As shown in Figure 4A, the molecular mass of untreated Gn-GMCSF and Gc are 72 kD and 61 kD, respectively. In contrast, in the corresponding PNGase F-treated samples, the Gn-GMCSF and Gc molecular mass are 70 kD and 56 kD, respectively. Therefore, both the chimeric Gn-GMCSF and Gc are glycosylated in the ER post-translationally (sensitive to PNGase F) and are consistent with their estimated molecular mass.

As shown in Figure 4B, the chimeric Gn-GMCSF is Endo H-sensitive, and the molecular mass of the Endo H-digested Gn-GMCSF is identical (70 kD) to the PNGase F-digested protein. Therefore, Gn-GMCSF was not processed in the Golgi. However, the chimeric Gc is Endo H-resistant; its molecular mass remained the same (61 kD) after the Endo H digestion. Therefore, the chimeric Gc was processed in the Golgi. These results are discussed further (Section 4) in light of the chimeric Gn sequence lacking the transmembrane domain sequence, which is deleted in the chimeric Gn-GMCSF.

#### 3.1.3. The BoHV-1qmv Sub-RVFV Vaccine Virus-Expressed Gn-GMCSF and Gc Proteins Form the Gn-Gc Complex

RVFV Gn and Gc form complex co-translationally upon proteolytic cleavage. In our chimeric Gn-GMCSF-FLAG-P2A-Gc-V5 gene cassette, P2A was incorporated to be cleaved co-translationally, which precedes the Gn–Gc complex formation [4]. To determine whether the chimeric Gn-GMCSF-FLAG and Gc-V5 form complex in BoHV-1qmv Sub-RVFV-infected cells, we performed reciprocal immunoprecipitation using anti-FLAG- and anti-V5-specific mAbs, followed by immunoblotting with anti-Gc or anti-Gn rabbit polyclonal antibodies to identify the co-immunoprecipitated proteins. As shown in Figure 5A, both the Gn and Gc are co-immunoprecipitated by the anti-FLAG or anti-V5 mAbs and are also recognized by their corresponding anti-Gn and anti-Gc-specific rabbit antibodies. Further, we immunoprecipitated the BoHV-1qmv Sub-RVFV with rabbit anti-Gn-specific antibody and immunoblotted with anti-FLAG and anti-V5 antibodies to validate the coimmunoprecipitation results by another way. As shown in Figure 5B the Gc is coimmunoprecipitated by the Gn-specific antibody. Therefore, as expected, the chimeric Gn-GMCSF-FLAG protein formed a complex with the chimeric Gc-V5 expressed by the BoHV-1qmv Sub-RVFV.

#### 3.1.4. Like the BoHV-1 wt, the BoHV-1qmv Sub-RVFV Vaccine Virus Replicated with a Similar Kinetics and Virus Yield in MDBK Cells but Produced Smaller Plaques

Two independent assays were performed to determine the plaque sizes and one-step growth kinetics of the BoHV-1qmv Sub-RVFV virus relative to the BoHV-1 wt virus. As shown in Figure 6A,B, the BoHV-1qmv Sub-RVFV produced significantly smaller plaques (approximately 68% reduction in plaque size) than the BoHV-1 wt virus. However, the one-step growth kinetics and virus yield of the BoHV-1qmv Sub-RVFV virus was similar to the BoHV-1 wt. (Figure 6C).

### 3.2. The BoHV-1qmv Sub-RVFV Is Stable in Expressing Gn-GMCSF and Gc Proteins after Ten Serial In Vitro Passages in Cell Culture

The stability of the BoHV-1qmv Sub-RVFV was determined by nucleotide sequencing and the expression of the chimeric Gn-GMCSF and Gc proteins after ten serial passages in cell culture. As shown in Figure 7, the FLAG- and V5-tagged, chimeric Gn-GMCSF and Gc, respectively, are unchanged between passages 1 and 10.

### 3.3. The BoHV-1qmv Sub-RVFV (Vaccine Virus) Is Highly Attenuated and Safe in Immunized Calves

For both the BoHV-1qmv Sub-RVFV and the parental BoHV-1qmv (parental vector virus) IN/subcutaneous inoculation, the calves were normal clinically.

### 3.4. Nasal Virus Shedding Following IN/Subcutaneous Immunization

BoHV-1qmv Sub-RVFV (vaccine virus) replication in the nasal epithelium was determined by virus isolation–titration and BoHV-1-specific qPCR assays of the nasal swab samples. Based on the virus plaque assay titers (Figure 8A, Appendix A) and BoHV-1 genome copies (Figure 8B, Appendix A), both BoHV-1qmv (parental virus vector) and BoHV-1qmv Sub-RVFV vaccine viruses replicated in the nasal epithelium with very similar efficiency and shed at least for five days in the nasal secretions of immunized calves.

### 3.5. A Single Vaccination Dose of BoHV-1qmv Sub-RVFV Induced a Robust (Vector-Specific) and Moderate (RVFV-Specific) Levels of Serum-Neutralizing (SN) Antibody Response in the Immunized Calves

At the time of immunization (0 dpv), calves in both groups had less than 4 (or 1) BoHV-1-specific serum-neutralizing (SN) antibody titers, which are considered negative. However, as early as 7 dpv, the mean SN titers in calves rose approx. ten-fold or more in both groups: 14.98 (BoHV-1qmv) and 11.93 (BoHV-1qmv Sub-RVFV), respectively (Figure 9A; Appendix A). By 21 dpv, the vector-specific (BoHV-1) SN antibody levels rose to 64 (a 60-fold increase) and 36 (a 35-fold increase), for the BoHV-1qmv and BoHV-1 qmv Sub-RVFV vaccine groups, respectively (Figure 9A).

Similarly, by 7 dpv, the mean RVFV (MP12 strain)-specific SN antibody titers in the prototype (BoHV-1qmv Sub-RVFV) vaccinated group rose to approx. 6.46 (Figure 9B; Appendix A). At 21 dpv, MP12-specific SN antibody titer rose to 17 (a 17-fold increase to that of 0 dpv), and at euthanasia (33 dpv), the RVFV-specific mean SN titer dipped only slightly to 14. In contrast, BoHV-1qmv (vector)-inoculated calves (negative control) were negative for MP12-specific SN titers until euthanasia (Figure 9B; Appendix A).

### 3.6. RVFV-Specific CMI Response in the BoHV-1qmv Sub-RVFV Immunized Calves

To determine whether BoHV-1qmv Sub-RVFV-vaccinated calves also generated cellular immune (CMI) response, we investigated IFN–γ responses in the PBMCs collected on 0, 14, and 21 dpv against RVFV upon stimulation, in vitro, with the heat-inactivated RVFV MP12 strain antigen. As depicted in Figure 10 and Appendix A, on 14 and 21 dpv, the PBMCs collected from the BoHV-1qmv Sub-RVFV-vaccinated calves induced a 8.34-fold and 3.63- fold increase in the levels of IFN–γ mRNA transcript relative to the PBMCs collected from BoHV-1qmv vector-inoculated calves, which remained negative and had no changes/fold changes in IFN–γ mRNA expression upon stimulation by RVFV antigen.

## 4. Discussion

The primary host of RVFV is livestock (cattle, sheep, and goats), which provides two critical ecological links between the primary vector, the *Aedes* sp., the floodwater mosquito, and the human population. First, livestock infected by the bite from transovarially infected *Aedes* sp. mosquitoes rapidly develop high viremias, allowing for the spillover of RVFV into secondary vectors (*Culex* and *Anopheles* sp., mosquitoes) that are more likely to feed on humans [30]. Second, the high viral loads found in livestock are also a significant risk factor for human infection by direct contact with contaminated blood, tissues, and aborted fetal materials [31]. Veterinarians, farm workers, and other health personnel are at high risk of infection from direct contact with infected animals. Many historical outbreaks of RVF disease in Africa were initially detected because of illnesses among these workers. The RVFV can be devastating among livestock, lethal in humans, and has widespread economic effects. Consequently, RVFV is uniquely suited for a one-health approach to prevent livestock and human diseases through animal vaccination.

In this study, we have developed a BoHV-1qmv-vectored subunit vaccine against RVFV (BoHV-1qmv sub-RVFV) and determined its safety and protective humoral and cellular immunogenicity in calves. Notably, the results showed that the chimeric subunit RVFV vaccine antigens, the Gn-swine GMCSF (fused), and Gc sequences, expressed as a polyprotein in BoHV-1qmv sub-RVFV-infected cells in culture, was cleaved proteolytically at the self-cleaving 2a peptide incorporated between the two chimeric proteins. The resulting Gn-GMCSF and Gc formed a non-covalently linked complex, as in the RVFV vaccine strain MP-12 case [4]. Further, we determined that the BoHV-1qmv Sub-RVFV is stable, highly attenuated, and safe for calves. BoHV-1qmv sub-RVFV-vaccinated calves generated RVFV-MP12 strain-specific neutralizing antibody and cellular immune responses.

Our results showed that both the chimeric Gn-GMCSF and Gc expressed in the BoHV-1qmv sub-RVFV-infected cells were glycosylated and formed heterodimer complex in ER. However, only the chimeric Gc glycoprotein and not the Gn-GMCSF became Endo H-resistant or was processed in the Golgi. The RVFV Gn has a 48-aa long Golgi localization signal (residues 585–632), and the Gc has an ER retention motif (aa residues 1194–1197) [4,5]. Therefore, Gc was expressed individually in transfected cells by Gc plasmid constructs localized in the ER, but when Gn was expressed separately, it localized in the Golgi and plasma membranes. In contrast, when Gc and Gn were co-expressed together, they formed heterodimer complex and localized in the Golgi [4,32]. Consistent with these parameters, we demonstrated that the chimeric Gn-GMCSF and Gc were co-immunoprecipitated because they formed a heterodimer complex, but only the Gc acquired Endo H resistance, confirming that Gc was processed further in the Golgi; however, the Gn-GMCSF unexpectedly remained Endo H-sensitive. In our chimeric Gn-GMCSF design, we deliberately omitted the transmembrane domain of the Gn (aa residues 582–604). Therefore, the first 20 residues of the Gn Golgi localization signal were deleted, leaving the remaining 28 aa of the Golgi-localization signal domain intact. Since the chimeric Endo H-resistant Gc was co-immunoprecipitated with the Gn-specific antibody, it was the indirect proof that chimeric Gn-GMCSF was not secreted but was instead localized in the Golgi with the Gc containing its transmembrane sequence intact. Yet, the Gn-GMCSF was Endo H-sensitive (unprocessed in the Golgi). Nevertheless, the chimeric Gn-GMCSF-Gc formed a complex, indicating that, conformationally, the chimeric heterodimer formation was unaffected. Therefore, one reason why the chimeric Gn was not processed could be that the fusion of the GMCSF with the Gn influenced the access of the enzymes required for trimming the high mannose sugars and the addition of the complex sugars to the single glycosylation site of the Gn [29].

For vaccination with BoHV-1qmv Sub-RVFV, we used both intranasal and subcutaneous routes. Intranasal vaccination is not affected by pre-existing antibodies against BoHV-1, and it also induces mucosal immunity. The subcutaneous route vaccination stimulates immune cells, including dendritic cells. Our vaccine design included the incorporation of GMCSF with the RVFV Gn as a secreted fusion protein. The rationale is that GMCSF is an attractive adjuvant for live virus-vectored vaccines [23] because of its ability to recruit antigen-presenting cells to the site of infected cells and stimulate the maturation of dendritic cells.

Although the protective efficacy of BoHV-1qmv Sub-RVFV against virulent RVFV challenge has yet to be tested, our results clearly showed that BoHV-1qmv Sub-RVFV is safe and induced optimal neutralizing antibody and CMI responses against RVFV in calves. Based on the results, we are optimistic about the BoHV-1qmv Sub-RVFV vaccines’ protective efficacy against challenges with virulent RVFV.

Our data also demonstrated that the BoHV-1qmv Sub-RVFV virus retains genomic stability for the RVFV Gn and Gc chimeric genes after ten passages in cell culture and a single passage in calves. Our data also revealed that the BoHV-1qmv Sub-RVFV does not shed in nostrils upon dexamethasone-induced latency-reactivation (Pavulraj and Chowdhury, unpublished data, manuscript in preparation). Therefore, the cattle population would have no vaccine virus transmission and circulation risk.

The live-attenuated RVFV MP12 vaccine retains some residual virulence because it induced abortion in 4% of ewes and teratogenic effects in 14% of newborn lambs [5,12], and the vaccine has the potential to regain virulence or revert back to wt by reassortment [14,15]. Several live virally vectored subunit-RVFV vaccines, using replication-competent, i.e., Newcastle disease virus [16] and Capripox virus [17], and replication-defective adenovirus [18,19], and vaccinia virus Ankara (MVA) [20], have been developed. Further, baculovirus-expressed RVFV antigen-based subunit vaccines [21] and RVF virus-like particles vaccines [22] have also been developed and tested for vaccine efficacy against RVFV. However, each has its own deficiency concerning vaccine efficacy, safety, or booster requirements [13]. Therefore, the availability of a safe, replication-competent BoHV-1qmv Sub-RVFV vaccine requiring only a single dose to generate protective immunity and prevent RVFV outbreaks in the livestock and human population of the affected region will be exceedingly valuable.

## 5. Conclusions

Our vaccine immunogenicity study revealed that a single dose of BoHV-1qmv-vectored subunit RVFV vaccination most likely induces protective humoral and cellular immune responses in calves, which will be validated by a future vaccination-challenge experiment.

## Figures and Tables

**Figure 1 viruses-15-02183-f001:**
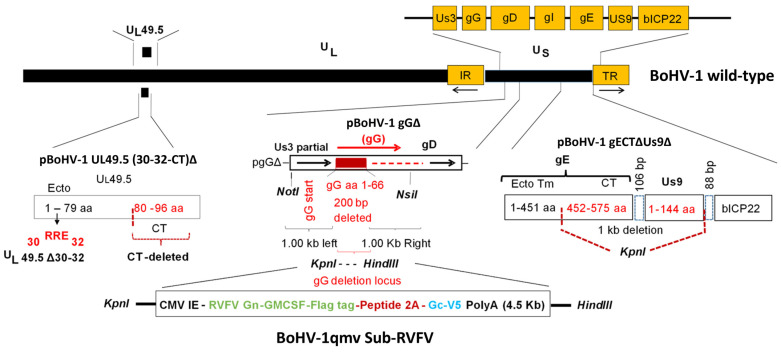
Schematic of bovine herpesvirus virus type 1 (BoHV-1) genome showing the UL49.5, glycoprotein G (gG), gE cytoplasmic tail (gECT), and US9-deletion loci and pBoHV-1gECT-∆Us9∆, pBoHV-1gG∆, and pBoHV-1UL49.5 (30-32-CT) ∆ used to construct the BoHV-1qmv vector. Also shown is the chimeric RVFV glycoprotein Gn-GMCSF fusion (Gn+), p2A, and Gc sequences insertion into the gG-deletion locus of the BoHV-1qmv genome, resulting in the BoHV-1qmv Sub-RVFV vaccine.

**Figure 2 viruses-15-02183-f002:**
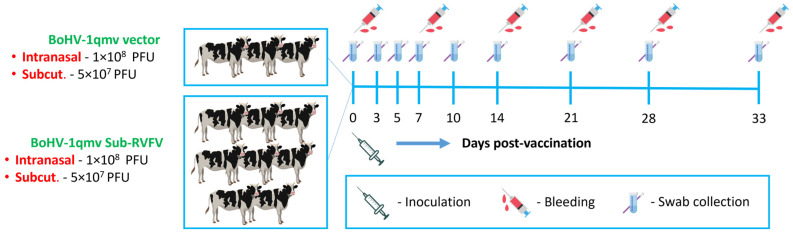
Schematic showing the immunization and sample collection scheme for the calf experiment. Intranasal—intranasal inoculation; Subcut.—Subcutaneous injection; PFU—plaque forming units.

**Figure 3 viruses-15-02183-f003:**
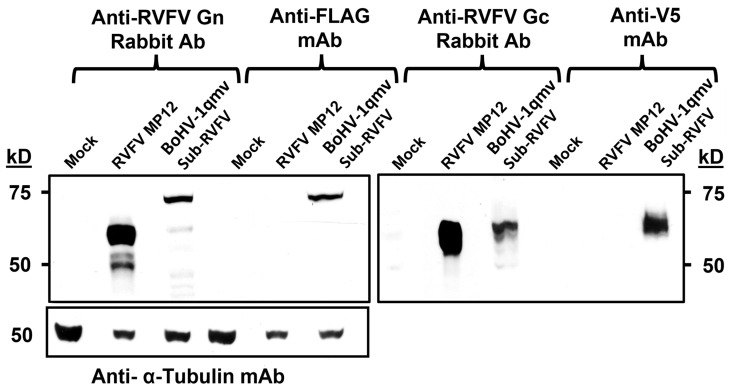
Immunoblot analysis of the BoHV-1qmv Sub-RVFV expressing chimeric RVFV Gn-GMCSF and RVFV Gc proteins using rabbit anti-RVFV Gn (left panel), anti-FLAG monoclonal antibody (mAbs) (middle left panel), rabbit anti-RVFV Gc (middle right panel), and anti-V5 mAbs (right panel), respectively. The RVFV Gn- and Gc-specific protein bands were absent in the mock-infected KOP-R cell lysates.

**Figure 4 viruses-15-02183-f004:**
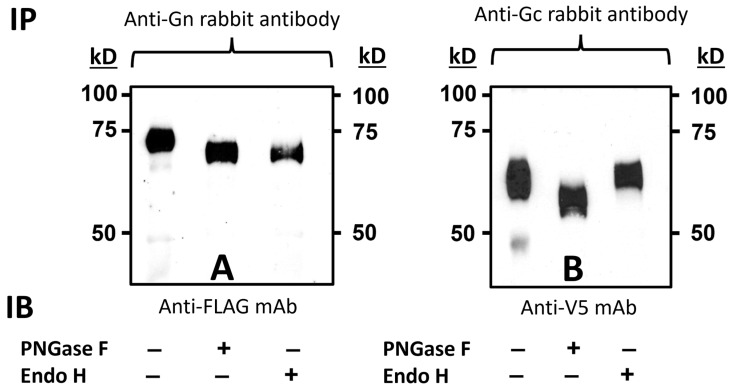
Immunoblot analysis of the BoHV-1qmv Sub-RVFV-expressed chimeric Gn-GMCSF and Gc proteins for glycosylation. KOP-R cells were infected with the BoHV-1qmv Sub-RVFV. Infected cell lysates were immunoprecipitated with either (**A**) RVFV Gn- or (**B**) RVFV Gc-specific antibodies. Immunoprecipitated proteins were either untreated (−) or treated with PNGase F (+) or Endo H (+) and subjected to SDS-PAGE and immunoblotting with FLAG tag (Gn-specific) or V5 tag (Gc-specific) antibodies.

**Figure 5 viruses-15-02183-f005:**
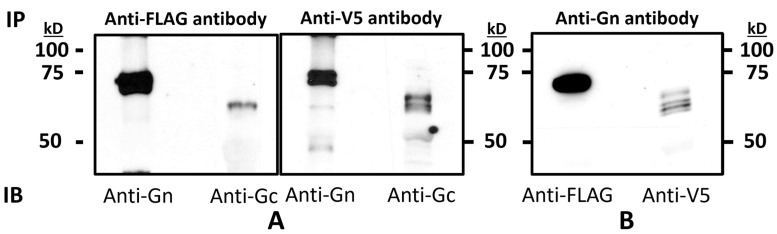
The BoHV-1qmv Sub-RVFV vaccine virus-expressed Gn-GMCSF and Gc proteins form the Gn–Gc complex in the infected cells in vitro. KOP-R cells were infected with the BoHV-1qmv Sub-RVFV. Infected cell lysates were immunoprecipitated with (**A**) FLAG tag- or V5 tag-specific antibodies followed by immunoblotting with RVFV Gc- or Gn-specific rabbit polyclonal antibodies to identify the co-immunoprecipitated proteins. (**B**) Similarly, the RVFV Gn-immunoprecipitated proteins were immunoblotted with FLAG tag- or V5 tag-specific antibodies to recognize the co-immunoprecipitated proteins.

**Figure 6 viruses-15-02183-f006:**
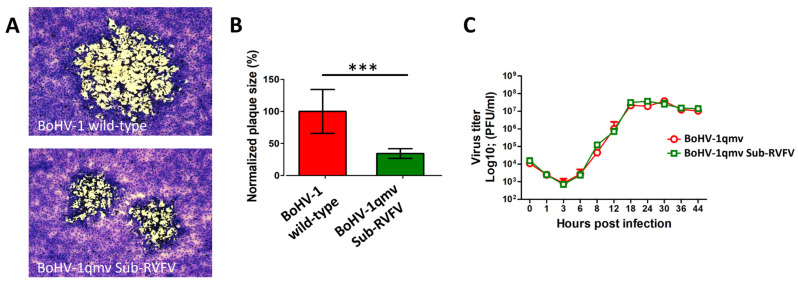
In vitro characterization of BoHV–1qmv Sub-RVFV. (**A**,**B**) Plaque size analysis of BoHV-1qmv Sub-RVFV compared to that of BoHV–1 wt. Shown are the pictures of areas containing representative plaques of each virus. The bar graph shows the average plaque size of at least 50 plaques with SD (*** *p* < 0.001). (**C**) One–step growth analysis of BoHV-1qmv Sub-RVFV compared with the BoHV–1 wt.

**Figure 7 viruses-15-02183-f007:**
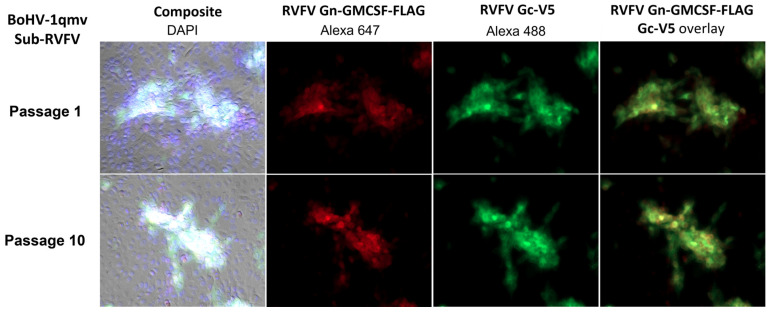
Indirect immunofluorescence assay (IIFA) to determine the stability of BoHV-1qmv Sub-RVFV. MDBK cells were infected with BoHV-1qmv Sub-RVFV from either passage 1 or passage 10. At 24 h post-infection, cells were fixed and IIFA were performed using rabbit anti-FLAG or mouse anti-V5 as primary antibodies and donkey anti-rabbit IgG Alexa Fluor 647 or donkey anti-mouse IgG Alexa fluor 488 as secondary antibodies. Chimeric protein expressions were indicated by bright far-red fluorescent signals for Gn-GMCSF-FLAG and bright apple-green fluorescent signals for Gc-V5 (Magnifications 200×).

**Figure 8 viruses-15-02183-f008:**
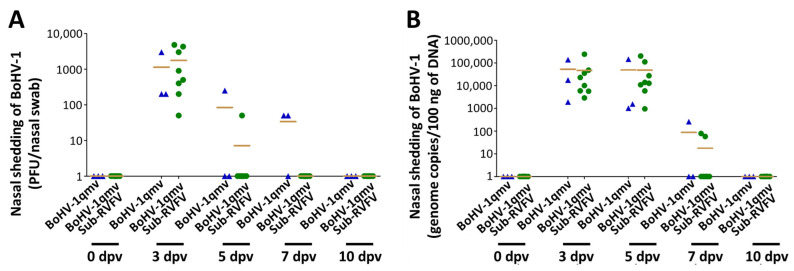
Nasal shedding of both BoHV-1qmv vector (▲) and BoHV-1qmv Sub-RVFV (
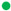
) in immunized calves assessed by qPCR and virus isolation. (**A**) After the inoculation into the KOP-R cells, the virus was isolated from each animal’s nasal swab and titrated in confluent KOP-R cells by plaque assay. Shown are the virus titers in the plaque-forming unit/mL of the nasal swab. (**B**) DNA was isolated from nasal swabs following inoculation and BoHV-1-qPCR was performed. The mean copy numbers of the BoHV-1 genome are shown in 100 ng of total DNA. PFU/mL; dpv—days post-vaccination.

**Figure 9 viruses-15-02183-f009:**
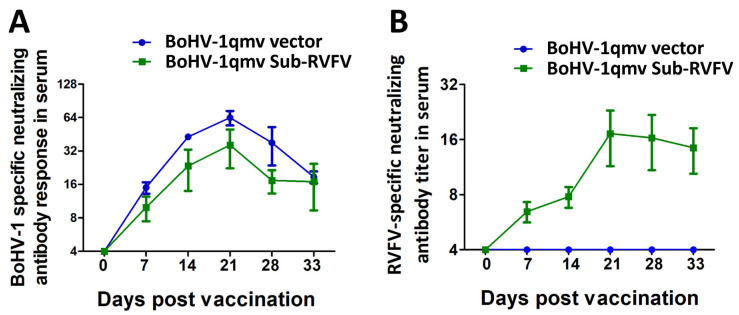
BoHV-1 and RVFV-specific neutralizing antibody titer in serum. BoHV-1- and RVFV-specific serum-neutralizing (SN) antibody titer developed in calves after BoHV-1qmv vector and BoHV-1qmv Sub-RVFV vaccination. (**A**) BoHV-1-specific SN titers. The data represent the mean + standard deviation. (**B**) RVFV-specific SN antibody titer following BoHV-1qmv vector and BoHV-1qmv Sub-RVFV immunization. The dot plot graph shows each animal’s mean values and individual titer with standard deviation.

**Figure 10 viruses-15-02183-f010:**
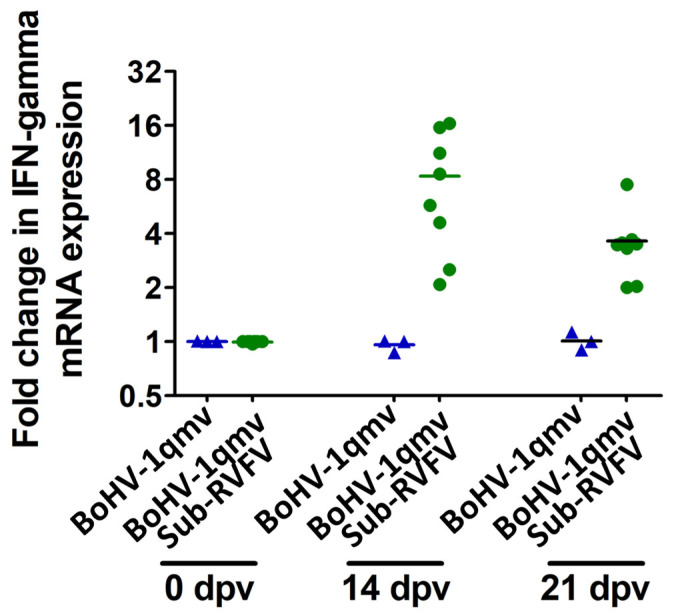
Fold-change in pre-vaccination and post-vaccination RVFV-specific interferon-gamma (IFN–γ) mRNA transcript expression upon in vitro stimulation by RVFV MP12 strain antigen.

**Table 1 viruses-15-02183-t001:** Amino acid sequences used to produce rabbit polyclonal anti-RVFV peptide-based antibodies.

Gene	Peptides	Amino Acid Sequence
**RVFV Gn**	Peptide 1	RNRPGKGHNYIDGMTQEDAT-Cys
	Peptide 2	SQCPKIGGHGSKK
	Peptide 3	ECTAQYANAYCSHAN
**RVFV Gc**	Peptide 1	Cys-VSSELSCREGQSYWTG
	Peptide 2	Cys-RNDKTFAASKGNRGVQAFSK
	Peptide 3	VLPSENGTKDQCQI

**Table 2 viruses-15-02183-t002:** Primers and probe sequences used for the qPCR analysis.

Primer/Probe/ds-Gblock		Sequence
**BoHV-1 major capsid protein**	Forward	5′-tttggaggccctagagaagc-3′
	Reverse	5′-aaacgtcaggtccatgttgc-3′
	Probe	5′Fam-cgggtgccctacccgctggt-3′Tamra
	ds-gblock	5′-ccgttggggaccggctagtgtttttggaggccctagagaagcgcgtgtaccaggccacgcgggtg ccctacccgctggtaggcaacatggacctgacgtttgtcatgccgctggggctgtacaaa-3′
**Bovine Glyceraldehyde 3-phosphate dehydrogenase (GAPDH)**	Forward	5′-catgaccactttggcatcgt-3′
	Reverse	5′-ccatccacagtcttctgggt-3′
	Probe	5’fam-accactgtccacgccatcactgcc-3’tamra
	ds-gblock	5′-ggcccccctggccaaggtcatccatgaccactttggcatcgtggagggacttatgaccac tgtccacgccatcactgccacccagaagactgtggatggcccctccgggaagctgtggcgt-3′
**Bovine interferon gamma**	Forward	5′-ccaggtcattcaaaggagca-3′
	Reverse	5′-tgcagatcatccaccggaat-3′
	Probe	5′Fam-tgaagtcctccagtttctcagagctgcc-3′Tamra
	ds-gblock	5′-cctcaaagataaccaggtcattcaaaggagcatggatatcatcaagcaagacatgtttcagaagttcttgaatgg Cagctctgagaaactggaggacttcaaaaagctgattcaaattccggtggatgatctgcagatccagcgcaa-3′

## Data Availability

All the available data are included in the manuscript.

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
