# Peer review of "A Novel Quadruple Gene-Deleted BoHV-1-Vectored RVFV Subunit Vaccine Induces Humoral and Cell-Mediated Immune Response against Rift Valley Fever in Calves"

_viruses, 2023, doi:10.3390/v15112183_

Round 1
Reviewer 1 Report
Comments and Suggestions for Authors
Dear editor,
thank you for giving me the chance to review Manuscript ID viruses-2667485 entitled “A Novel Quadruple Gene Deleted BoHV-1-Vectored RVFV 2 Subunit Vaccine Induces Humoral, and Cell-mediated Immune 3 Response Against Rift Valley Fever in Calves” by Selvaraj Pavulraj et al. for Viruses. The workgroup of Shafiqul I. Chowdhury has well known experience in the field of molecular biology of bovine and porcine herpesviruses and vaccine development and contributed significantly to the field with their previous works.
The main topic of the current manuscript is a novel quadruple gene deleted BoHV-1-vectored RVFV subunit vaccine. The authors show that a single dose of this vaccine induces humoral and cellular immune responses in calves. This results are highly interesting for other researchers in the field, since better vaccines for RVFV are needed, and the described vaccine has the potential to prevent RVFV outbreaks in both livestock and human populations in affected regions with a better safety profile than currenty available vaccines. However, the main drawback ist the lack of challenge experiments showing a protective effect against virulent RVFV strains in vivo. This has to be done in future studies.
In summary, this is a well written and highly interesting manuscript concerning zoonotic Rift Valley fever. I recommend acceptance of this manuscript for Viruses after minor modifications.
Points of criticism:
1. The manuscript describes the induction of humoral and cellular immune responses in calves with a BoHV-1-vectored RVFV subunit vaccine. However, it does not show protection, since challenge experments have not been performed. Therefore, the manuscript is massively overinterpreting the results in multiple instances. The authors should interpret their results more cautionesly and rephrase all statements like abstract page 1 lines 32-34, suggesting “protective” immune responses or conclusions that this vaccine “can be used as an efficacious subunit vaccine against RVFV in cattle “. This can only be concluded after challenge experiments with virulent RVFV strains.
2. Is the co-expression of GM-CSF really necessary and safe? I am in fear, that the co-expression of this protein in a slightly altered conformation might induce neoepitopes followed by a higher risk for autoimmune-mediated disease in the vaccinated bovines. Anecdotally, starting in 2006 we observed a cluster of BVD-vaccine (PregSure® BVD (Pfizer GmbH, Berlin, Deutschland)-induced bovine neonatal pancytopenia in Germany (Buck et al 2011 Berl Munch Tierarztl Wochenschr 124(7-8):329-336). To my knowledge, the epitope was later found to be MHC 1 (Benedictus et al. 2017 Expert Rev Vaccines 16(1):65-71). However, couldn’t immune-mediated depletion of a factor involved in hematopoietic cell induction like GM-CSF could lead to a comparable syndrome of pancytopenia? Increasing the risk for autoimmunity only slightly will not be obvious in a study with a small N, like this one. Therefore, I suggest to discuss this issue with more detail and formulate the statements on safety more cautiously.
3. Your construct contains parts of Gn and Gc correct? Why did you use parts of two different viral antigens? Would one or parts of one be enough? What is the most effective, minimal epitope in immunity against RVFV. Please include your rationale for epitope selection in the manuscript.
Reviewer 2 Report
Comments and Suggestions for Authors
The manuscript by Pavulraj et al., describes the deveopment of a Gene Deleted BoHV-1-Vectored RVFV Subunit Vaccine that was shown to induce humoral, and cell-mediated immune response against Rift Valley fever in calves.
specific comments:
1. the electron microscopy experiments are inconclusive. the VLPs that are shown in blue arrows need to be identified using gold-labelled antibodies against RVFV Gn/Gc for confirmation. A lot of membrane structures resemble VLP and can be wrongly identified as VLPs. Alternatively, the infected cell culture supernatant can be centrifuged to pellet the VLPs that can be then visualized by EM and tha same confirmed by western blotting.
2. Were there any mock infected but in-contact animals that were tested to see if the vector virus particles were able to transmit to uninfected animals.
Comments on the Quality of English Language
few minor grammatical and spelling corrections needed
Reviewer 3 Report
Comments and Suggestions for Authors
Rift Valley Fever Virus (RVFV) is a zoonotic disease that causes hemorrhagic fever in livestock and other animals as well as humans, and its prevention is an urgent public health issue. The authors used a mutant strain of bovine herpesvirus type 1 (BoHV-1) (BoHV-1qmv), which lacks virulence and immunosuppressive properties, as a viral vector to generate an RVFV envelope protein Gn and Gc expressing virus (BoHV-1qmv Sub-RVFV) was created. BoHV-1qmv Sub-RVFV replicated to high titers in cell culture and produced RVFV-like particles. Intranasal and subcutaneous inoculation of BoHV-1qmv Sub-RVFV to calves showed safety similar to that of the parent strain and induced specific neutralizing antibodies against the RVFV vaccine strain MP12. Additionally, stimulation of peripheral blood mononuclear cells isolated from calves vaccinated with BoHV-1qmv Sub-RVFV with heat-inactivated MP12 antigen increased by 6 times interferon-gamma mRNA levels compared to unvaccinated control calves. These results demonstrate that a single dose of BoHV-1qmv Sub-RVFV induces protective humoral and cellular immune responses in calves. The research methods are well thought out and clearly described. Overall, the content is very interesting for readers.
[Minor point]
1. Line 146: Wouldn’t “5’ -3’” be better “5’ to 3’”?
2. Lines 206-207, 252-253, 381-382, 406-407, 443-444, 457-458, 511-512: One line of space is required.
3. Line 330: Isn’t “Gn_-GMCSF” “Gn-GMCSF”?
4. Line 415: Isn’t “Sub-RVFV_-infected” “Sub-RVFV-infected”?
5. Line 496: Isn’t “as _7 dpv” “as 7 dpv”?
